# Model-Based Roentgen Stereophotogrammetric Analysis to Monitor the Head–Taper Junction in Total Hip Arthroplasty in Vivo—And They Do Move

**DOI:** 10.3390/ma13071543

**Published:** 2020-03-27

**Authors:** Jing Xu, Robert Sonntag, J. Philippe Kretzer, Dominic Taylor, Raimund Forst, Frank Seehaus

**Affiliations:** 1Department of Orthopaedic Surgery, Faculty of Medicine, University of Erlangen-Nürnberg, 91054 Erlangen, Germany; 2Laboratory of Biomechanics and Implant Research, Clinic for Orthopedics and Trauma Surgery, Center for Orthopedics, Trauma Surgery and Spinal Cord Injury, Heidelberg University Hospital, 69118 Heidelberg, Germany

**Keywords:** Roentgen stereophotogrammetric analysis, hip arthroplasty, elementary geometrical shape model, interchangeability, head–taper junction, migration

## Abstract

Model-based Roentgen stereophotogrammetric analysis (RSA) using elementary geometrical shape (EGS) models allows migration measurement of implants without the necessity of additional attached implant markers. The aims of this study were: (*i*) to assess the possibility of measuring potential head–taper movement in THA in vivo using model-based RSA and (*ii*) to prove the validity of measured head–taper migration data in vitro and in vivo. From a previous RSA study with a 10 years follow-up, retrospectively for *n* = 45 patients head–taper migration was calculated as the relative migration between femoral ball head and taper of the femoral stem using model-based RSA. A head–taper migration of 0.026 mm/year can be detected with available RSA technology. In vitro validation showed a total migration of 268 ± 11 µm along the taper axis in a similar range to what has been reported using the RSA method. In vivo, a proof for interchangeable applicability of model-based RSA (EGS) and standard marker-based RSA methods was indicated by a significant deviation within the migration result after 12-month follow-up for all translation measurements, which was significantly correlated to the measured head–taper migration (r from 0.40 to 0.67; *p* < 0.05). The results identified that model-based RSA (EGS) could be used to detect head–taper migration in vivo and the measured movement could be validated in vitro and in vivo as well. Those findings supported the possibility of applying RSA for helping evaluate the head–taper corrosion related failure (trunnionosis).

## 1. Introduction

Modern total hip arthroplasty (THA), referred as the “surgery of the century”, shows survival rates of >95% after 10 years and of >80% after 25 years [1]. One of the top five clinical challenges within THA is aseptic loosening [2], which presents not only a question of wear [2]. However, implant-to-bone movements, termed as migration, are considered as one the most important mechanical factors for aseptic loosening [3]. Marker-based Roentgen stereophotogrammetric analysis (RSA) presents the common gold standard to detect in vivo implant migration [4] (Figure 1A). However, additional implant marking could be associated with increased production/recertification costs for implant manufacturer as well as with an increased risk of cement cracking [5]. To overcome these disadvantages of standard marker-based RSA, a model-based RSA method was introduced, using 3D surface models of the implant to match models projection contour on the implant contour with radiographic RSA image pairs to calculate the 3D position of the implant [5]. Sourcing of the required surface models can be achieved by reverse engineering (RE) or by computer aided design (CAD) method. Interchangeability of model-based RSA approach using CAD/RE models to standard marker-based RSA could be verified [5,6,7]. Model-based RSA using elementary geometrical shape (EGS) models is a further RSA approach to measure migration of THA components [8]. The EGS approach using spheres and cylinders or hemispheres, to represent femoral head or stem (Figure 1B) or acetabular cup component, respectively. Bone markers are still required.

In addition to articular wear, many clinical studies have indicated corrosion between the femoral ball head component and the stem-taper as one of the sources of THA failure [9,10,11,12]. This phenomenon is also called trunnionosis, which is defined as the wear and corrosion that occurs at the junction of the modular head–taper or neck-stem initialized by micromotion [13]. There are local corrosive processes, but also mechanically induced processes taking place at that interface over the lifetime of the implant. A subsequent head–taper migration respective to the seating of the ball head on the male stem-taper is multifactorial, including the implant design (e.g., technical specification of male/female tapers), the surgical technique (e.g., impaction force) or patient-related factors (e.g.,/ body mass index, activity) [14]. In addition, the geometry and topography of both taper parts, the female ball head and the male stem-taper, determine the local effects. In that context, it is important to know that even though many tapers are called the same, e.g., a 12/14 standard taper—they are far from being standardized and vary between manufacturers [15]. The same is true for the topography, which is mainly characterized for the male stem-taper.

According to previous research, trunnionosis can cause clinical symptoms such as adverse local tissue reactions (ALTRs), thigh pain and local swelling [16]. Instability and special gait patterns was also observed in some cases [17]. The overall incidence of trunnionosis of primary THA cases was reported at approximately 1–3% [18,19]. At present, laboratory investigations of trunnionosis are limited to the detection of the metal ion in serum and joint capsule. Magnetic Resonance Imaging (MRI) and ultrasound present good specificity and sensitivity on detecting pseudotumor and ALTRs [20,21]. Severe osteolysis caused by ALTRs can be viewed on plain radiographs. However the plain radiograph is insufficient for the detection of head–taper micromotions. Sultan et al. [14] stated that the true incidence rate of trunnionosis was underestimated as the release of metal debris can also cause osteolysis and aseptic loosening.

To the authors’ knowledge, no clinical measurement tools commonly exists to measure or monitor the possibility of head–taper migration in vivo. Model-based RSA or a combination with standard marker-based RSA offers the opportunity to measure a possible motion within the head–taper junction. However, this option is limited in application to hard-soft bearings only, while visibility of the ball head’s contour is necessary within the RSA radiographic pairs.

Therefore, the purpose of this study is (*i*) to assess the feasibility to detect a possible head–taper movement in THA in vivo using model-based RSA and (*ii*) to prove the validity of measured migration data. The authors hypothesize thus a measurement of head–taper migration (if available within the given clinical records) for hard-soft bearings is possible and thus the measured migration is valid.

## 2. Materials and Methods

Data presented was taken out of a previous RSA study [22] initiated in 1999 (ethical registration number: 1.077). It offers the opportunity to study long-term implant migration as well as to measure it by the both marker-based and model-based RSA (EGS) method, respectively.

### 2.1. Patient Cohort—Inclusion Criteria

All patients who were included had received a cemented femoral stem (Lubinus SP II, Waldemar Link GmbH, Hamburg, Germany) with the three visible additional attached tantalum markers and a cemented polyethylene acetabular cup (LINK^®^ IP Acetabular Cup, Waldemar Link GmbH, Hamburg, Germany) component. Each femoral stem has a caput–collum–diaphyseal angle of 126 degrees and is combined with a ceramic ball head (BIOLOX^®^*forte*, Ceram Tec GmbH, Plochingen, Germany) diameter of 28 mm. Three different sizes of head–neck length were used (S, M and L size, Table A1 and Table A2). In summary, *n* = 45 cases could be analyzed. To the authors’ knowledge, the full weight bearing of the THA treated hip was first allowed within the rehabilitation period.

### 2.2. Image Acqusition and Analysis

All patients underwent the first reference RSA examination during the first postoperative week and received follow-up RSA examinations at 1.5 months, 3 months, 6 months, 1 year, 2 years, 5 years, and 10 years. In all RSA follow-ups, the same equipment and parameters were used. Patients were positioned in the supine position on the X-ray table. The X-ray table is placed within the uniplanar RSA setup. The RSA setup consists of two X-ray sources, focused at the patient hip with an intersection angle of 40 degrees directly above the patient. Underneath the X-ray table, a calibration box (Umeå 43, RSA BioMedical Innovations AB, Umeå, Sweden) was placed, with both of the X-ray films in the lower level of the calibration box. The X-ray film-focus distance was 140 cm.

All resulting RSA image pairs were analysed using model-based RSA method (MBRSA 4.1, RSA Core, Leiden, The Netherlands). Migration was calculated based on a reference point, which is usually represented by the center of gravity of the rigid body. For image analysis, the RSA standard thresholds (calibration errors translation ≤ 0.05 mm, focus error ≤ 0.5 mm; condition number ≤ 100; rigid-body error ≤ 0.35 mm) were applied to verify the quality of the image calibration procedure [23,24].

Migration results were represented with respect to a global coordinate system defined by the calibration box: translation along the medial–lateral (x) and cranial–caudal (y) axes constitute in-plane motion, and translation along the anterior-posterior-axis (z) out-of-plane motion; rotation around the anterior-posterior axis (Rz) further described in-plane motion, and along the medial–lateral (Rx) and cranial–caudal axes (Ry), and out-of-plane motion, respectively.

### 2.3. Head–Taper Migration Measurements

Using a model-based RSA approach (Figure 2) the head–taper migration was calculated between the rigid body of the femoral stem component (defined by the three additional attached implant markers) and the ball head (represented as a spherical EGS model). Migration results were calculated according to the above-mentioned protocol and displayed by using the allocated coordinate system.

To take into consideration that the femoral ball head can move mechanically along the long axis of the taper, a new coordinate system based on the direction of the long axis of the taper and three implant markers was set up to measure head–taper migration (Figure 2).

The taper contour was matched with an EGS cone model to find the orientation of its central axis, which became the Y-axis of this new coordinate system. The X-axis of the new coordinate system was set orthogonal to Y-axis and parallel to a plane across three implant markers and pointing in a medial direction. The Z-axis of the new coordinate system was orthogonal to the X- and Y-axes and pointing in an anterior direction. Then the original head–taper migration results were transformed according this new coordinate system.

### 2.4. Experimental Verification of Head–Taper Migration

Based on a previous study analyzing the taper geometry and topography of different manufacturers [15], *n* = 5 CoCr sample tapers matched with the taper profile of the manufacturer of the cemented femoral stems used in the RSA study (Link GmbH, Hamburg, Germany). The geometry of the sample tapers has been verified by a coordinate measuring machine (Mahr Multisensor/MarVision MS 222; Mahr, Göttingen, Germany; accuracy: ±2.3 µm) and the topography by a tactile roughness measurement instrument (Perthometer M2; Mahr, Göttingen, Germany; accuracy: 12 nm). For the experimental investigation of head–taper migration, ceramic ball heads (BIOLOX^®^*delta*, Ceram Tec GmbH, Plochingen, Germany) of 28 cm diameter and head–neck length M have been used. The length of the head sample taper assembly has been measured after assembly by hand (0 kN) and after impaction at two load levels (2 kN and 4 kN) using a servo-hydraulic testing machine (MTS 858 Mini Bionix II; MTS, Eden Prairie, MN, USA).

### 2.5. Clinical Verification of Potential in Vivo Head–Taper Migrations—Proving Interchangeable Applicability of Marker and Model-Based RSA EGS Method

To prove data validity of the in vivo obtained migration results, a test of interchangeable applicability of measured marker-based and model-based RSA (EGS) is necessary. In summary *n* = 18 cases from the above mentioned study cohort were identified, for which a complete set of the RSA follow up examinations were available.

Each pair of RSA radiograph was analyzed three times (one week break for the observer is predefined and respected) by both marker-based and model-based RSA EGS approach according to a written protocol. Since both RSA methods calculate their implant-to-bone migration from the same image data sets, the results need to be identical. However, according applied reference points for migration detection, special attention has to be paid on (1) reference point correction and (2) EGS model generation. 

If the implant-to-bone migration including rotational motion, which is present within most of clinical situations, the resulting translational migration results can be variant if different reference points were used (Figure 3) [6]. To enable the comparison of migration results from marker- and model-based RSA (EGS) approaches, a reference point correction described by Hurschler et al. [6] was performed (Mathematica, Wolfram Research, Champaign, IL, USA). The reference point of model-based RSA (EGS) was corrected to the same position as in marker-based RSA with a subsequent recalculation of the translational results in all three axes (Figure 3).

In the model-based RSA (EGS) approach, the femoral stem component of a THA design was displayed by a cylinder and a sphere for the migration measurement (Figure 4A). Three virtual markers resulting out of the arrangement of both the EGS models. In comparing the marker-based RSA (where the reference point is calculated from three rigidly fixed tantalum markers on the femoral stem) with the model-based RSA (EGS) whose reference point could be variable, thus creating head–taper migration (Figure 4B). Due to the reference point variability (in the model-based RSA (EGS) approach) and its effect on the migration results: a head–taper migration can induce additional movement to the migration results.

The two above-mentioned details about reference point behavior, enable the clinical verification of measured head–taper migration. If a head–taper movement exists, the virtual markers and the reference point are changed respectively (from P_1_ to P_5_ and P_3_ to P_4_ as well as R_1_ to R_2_; Figure 4) whilst using model-based RSA approach for migration detection. This indicated a deviating rigid body representation in comparison to standard marker-based RSA, which use three additional fixed tantalum markers to represent a monobloc of the femoral stem (Figure 3B). A marker movement cannot occur, thus a change in the reference point is not possible.

### 2.6. Statistics 

All statistical analyses were carried out by using R (R Foundation, Vienna, Austria). Head–taper migration was presented by using descriptive statistics. Student’s *t*-test was performed to evaluate the significance of head–taper migration over the 10 years follow up period.

All experimental data is given as a mean and a standard deviation for the *n* = 5 head-stem samples tested.

To verify measured head–taper migration, interchangeable applicability of marker- and model-based RSA EGS approaches were analysed in six degrees of freedom using the Bland–Altman plots [25]. Bland–Altman plots were used as an unscaled method to calculate the mean inter method difference and limit of agreement (LoA) between the both of the RSA approaches. According to the literatures [26,27], the upper limit of RSA accuracy (0.5 mm for translation, 1.15 degrees for rotation) was used as the upper threshold of LoA to determine whether the two RSA methods are interchangeable or not. A one-sample student’s *t*-test (significance level = 0.05) was performed to evaluate the significance of mean difference between two RSA methods. The Pearson correlation coefficient was used to assess a linear correlation between intermethod difference and head–taper migration [28]. To test the working hypothesis, a test based on Fisher’s Z transformation [29] was used to establish that the true correlation coefficient is not equal to 0 with the significance level of 0.05.

## 3. Results

### 3.1. Head–Taper Migration

Migration of the femoral ball head is caused by a creeping embedment in the direction of the taper throughout the follow-up period. The majority of the migration was discovered to occur in the 6- to 4-month follow-up period (Figure 5B). A translation towards the taper along the longitudinal axis (y-axis) from 0.028 to 0.115 mm (*p* < 0.01) was measured. The migration speed slowed down in 24 to 120 month follow-up period.

Additionally a continuous migration perpendicular to the long axis of the taper and in a medial-posterior direction (x-axis translation: from 0.044 to 0.139 mm, *p* < 0.01; z-axis translation: from −0.109 to −0.279 mm, *p* < 0.05) was also observed in the 6 to 24 month period, and then tended to stabilize (Figure 5A,C). A mean head–taper migration of 0.026 mm per year, over the 10-year follow-up period was identified.

A subgroup analysis using head–neck length, indicated a similar migration pattern for all three groups. The L size group showed a trend of greater migration along medial-lateral (x-axis) direction and in the anterior-posterior direction (z-axis) than the other head–neck length groups (Figure 6). However, results of the student’s *t*-test showed that it was insufficient to demonstrate a significant difference between any two groups.

### 3.2. Experimental Verification of the Measured in Vivo Head–taper Movement 

Male taper geometry and topography have been shown to be comparable to what has been reported in the literature for the femoral stem investigated in this study (Table 1) [15]. As expected, migration has been shown to be largest after initial impaction at 2 kN (162 ± 6 µm) and smaller after subsequent impaction at 4 kN (106 ± 10 µm) (Table 2). In total, a head–taper migration of 268 ± 11 µm was measured during the experimental investigation.

### 3.3. Clincial Verification—Proof Interchangeability between Marker-Based RSA and Model-based RSA (EGS) Migration Results

A discrepancy between the marker and model-based RSA (EGS) approach, especially in cranial-caudal axis from 12 to 120 months is observable (Figure 7). Mean translation along cranial–caudal axis measured over all follow-ups (mean value from 3 to 120 months) by model-based RSA (EGS) was −0.083 mm, which was much greater than that measured by marker-based RSA (−0.021 mm).

The LoA between the both RSA methods were acceptable for in-plane migration (Figure A1), less than 0.04 ± 0.34 mm on translation measurement, and less than 0.02 ± 0.39 degrees on rotation measurement. For out-of-plane migration (Figure A2) LoA was greater than in-plane measurement. Especially on rotational measurement around cranial–caudal axis, the LoA between two methods was −0.47 ± 3.78 degrees, which was much greater than that of other directions (medial-lateral rotation: −0.01 ± 0.88 degrees, anterior–posterior translation: 0.02 ± 0.53 mm). 

After dividing all cases into two groups (3 to 6 months and 12 to 120 months), most of the measurements showed a similar range of the LoA between the two groups (Figure 8, Figure A3). Whereas, in all of translational measurements from 12 to 120 months, the mean difference between two RSA methods was found to have a significant bias from 0 (*p* < 0.05) (Figure 8B,D,F).

### 3.4. Correlation between Intermethod Difference and Head–Taper Migration

The intermethod difference and head–taper migration showed a strong linear correlation in all three axes with r ranging from 0.40 to 0.67 (*p* ≤ 0.05) and correlated most in the cranial–caudal axis (r = 0.67, *p* ≤ 0.01) (Figure 9). Analyses with each follow up points exhibited a strong correlation at most of the follow-up time points (Table A3).

## 4. Discussion

Series of clinical complications can be caused by head–taper corrosion, including ALTRs, pseudotumors and even the unacceptable gross taper failures [30]. Recently, MRI and ultrasound are regards as best methods for detecting ALTRs and pseudotumors caused by metal debris [20,21]. To the authors’ knowledge, there is no clinical measurement tool available to monitor the head–taper migration in vivo. For hard-soft bearings (MoP and CoP), model-based RSA method or a combination with the standard marker-based RSA represents an approach that offers this option, and could be a potential candidate for the evaluation of head–taper migration. Since this method uses rigid-body kinematics to determine the relative motion between two rigid bodies, an in vivo assessment of the head (using a spherical model of the ball head) and taper (using an EGS model of the stem component or additionally attached implant markers) junction would be enabled. However, this application requires the visibility of the femoral ball head contour within the resulting RSA radiographic pairs.

Head–taper migration could be detected using model-based RSA (EGS) method retrospectively from the available previous clinical RSA images. Measured migration rates within the first 6 months were low, while they increased continuously from 6 to 24 months post-op (along the taper: from 0.028 to 0.115 mm; perpendicular to the taper: from 0.118 to 0.312 mm). Migration slowed from 24 to 120 months. Therefore, the hypothesis (*i*), related to the in vivo assessment of head–taper movement in THA using model-based RSA approach could be accepted. Benefited from the good accuracy of RSA technology, it is possible to measure the submillimeter head–taper migration. Even for cases with stable implants (no case included in this study failed during the 10 years of follow-up period), model-based RSA (EGS) is able to detect the migration of head–taper junction within the necessity of additional implant or bone marking. Therefore, the presented results validated the possibility of RSA technology for head–taper migration measurement and enable a wider applicability. It is hoped that this will improve the ability to evaluate further couplings within THA next to investigated metal- ceramic coupling of head–taper stem.

The in vitro as well as an in vivo validation of measured head–taper migration supports the findings of retrospective analyses. The experimental data shows a migration along the taper axis in a similar range (up to approximately 250 µm) to what has been measured using the RSA approach. The discrepancy between the in vitro and in vivo data may be related to different starting references. While in the RSA group, the head had already been intra-operatively impacted and migration was measured over time with more and more loading of the patient, the experiments started with a nonimpacted status (manual assembly) and increased the impaction force without the investigation of any dynamic loading. However, the findings indicate that the migration measured by the RSA method is in a very reasonable range.

The clinical validation (in vivo) of measured head–taper migration could be confirmed by proving interchangeable applicability of model-based RSA (EGS) approach next to standard marker-based method. In the first 6 months follow up period, the results of individual measured migration by both RSA methods, supported and demonstrated the interchangeable applicability of two methods (Figure 8, Figure A3). The results of this study showed that the migration between investigated femoral stem and applied femoral ball head occurs mostly take part between 6 to 24 months post-operatively, which lead to the significant difference (*p* < 0.05) between the two RSA methods on all translational measurements after 12 months follow up (Figure 8). Before head–taper migration accelerated (between 3 and 6 months post-operation), the migration measurements with EGS model did not have significant difference compared to the gold standard marker-based RSA on cranial–caudal and anterior–posterior translation (*p* > 0.05). Migration and LoA from 12 to 120 months indicate a different behavior. The results of the correlation analysis showed that there was a strong correlation between head–taper migration and intermethod difference (r from 0.40 to 0.67; *p* < 0.05). As an additional factor, head–taper migration could introduce a systematic error to the migration measurement of hip stem prosthesis when using EGS model, or the continuous migration between head and taper can be regarded mistakenly as a continuous instability of the hip stem. If the head–taper migration was extremely large and leads to an obvious change of the relative marker position, the rigid body error would exceed the threshold of 0.35 mm and make the migration analysis meaningless in this situation. However, if the head–taper migration was relatively small, its contribution to the measurement error could be negligible. Hence, the results of EGS stem model need to be treated cautiously only when the head–taper migration was relatively large but the rigid body error was still within the threshold.

Based on this hypothesis, (*ii*) could be accepted. A measurement of head–taper migration for hard-soft bearings is possible and thus the measured migration is valid. However, the measured head–taper migration pattern can be different in other implant designs.

The presented results showed a migration along the taper axis with higher values after implantation and a reduced migration speed over time. This phenomenon can be explained as follows: Insufficient force applied during the impaction of the femoral head is one of the pathogenesis of head–taper migration [31]. According to the presented in vitro experiment, after the femoral head has been impacted with lower force (2 kN), increased force (4 kN) can cause further plastic deformation (seating) on the taper even under noncorrosive conditions. From this it is speculated that in the early post-operative period, the head–taper migration was dominated by reasons induced by mechanical/material deformation, and showed a relatively high speed. In the later period, the corrosion dominated and resulted in a relatively lower migration. However, exploring the causes of head–taper migration in each specific period is beyond the scope of this study, and need to be further confirmed by clinical retrieval studies. In general, high assembly forces generate the safest situation for the head–taper junction, resulting in the greatest resistance to relative motion over time [32].

In addition to the migration along the longitudinal axis of the taper, the femoral head prosthesis revealed also migration in other directions perpendicular to the taper. According to previous literature, the authors believe that this type of migration was caused by the fretting corrosion in-between the head–taper interface [33]. The periodic pressure during walking is transmitted from femoral head prosthesis to the stem prosthesis through the taper, resulting in a long-term oscillatory slip at the head–taper interface. The passive layer of material at the contact surface can be gradually destroyed by this mechanical action and then corrosion is accelerated [34,35]. This phenomenon can be much worse in metal-to-metal contact surfaces (e.g., titanium head vs CoCr taper). The released metal debris could induce ALTRs and aseptic lymphocytic vasculitis associated lesion [36]. The inflammatory response to the metal debris can also be related to the following bone resorption and aseptic loosening. While ceramic head prosthesis can slow down the taper corrosion speed and also reduce the released metal debris [37,38,39,40]. Besides the adverse reactions of metal debris, the material loss caused by corrosive and mechanical abrasion can destroy the force-locking of the head–taper junction, further exacerbating the wearing-out of taper material. The end phase of this process was gross stem-taper failure [30]. Monitoring of head–taper migration in context of corrosive processes is essential. It may be that corrosive processes change the location of head-stem-taper, which is generally proximal for a ceramic head. For a more distal contact over time, the risk of a ceramic head fracture may be increased [41].

Head–neck length is another factor that influences the corrosion speed [42]. A head with larger head neck length correspondingly has a smaller head–taper contact area, resulting in a stress concentration and more corrosion at the interface [32,43]. The results of this study showed that the large head neck length group had slightly more corrosion than the other two groups. However, concerning the aspect of head–neck length and interface corrosion, there was insufficient evidence to support the statistical difference. The limiting factor being the small sample size in each group.

## 5. Conclusions

Overall, RSA is a reliable method for measuring in vivo implant migration. The results of this study demonstrate that model-based RSA can be used to detect head–taper migration for hard-soft bearings in THA and could be a potential candidate for the clinical evaluation of trunnionosis. In addition, future diagnostic studies are essential for further understanding the effect measured by the RSA method. For patients with high risk factors, more detailed tests should be performed during the follow-ups to reduce the possibility of misdiagnosis in clinical.

Furthermore, clinical validation of measured head–taper migration could be confirmed, thus demonstrating the interchangeable applicability of model-based RSA (EGS) to standard marker-based RSA method. Observed deviation of the measurements between the two RSA methods from 12 to 120 months postoperatively may be explained by head–taper migration, which may confirm the validity of the findings. It has been statistically shown that head–taper migration was correlated to the deviation between methods. Thus, the results indicate that it is necessary to prove the head–taper migration during the entire study period using model-based RSA EGS approach, so that no misinterpretations are drawn from the resulting migration values happens.

## Figures and Tables

**Figure 1 materials-13-01543-f001:**
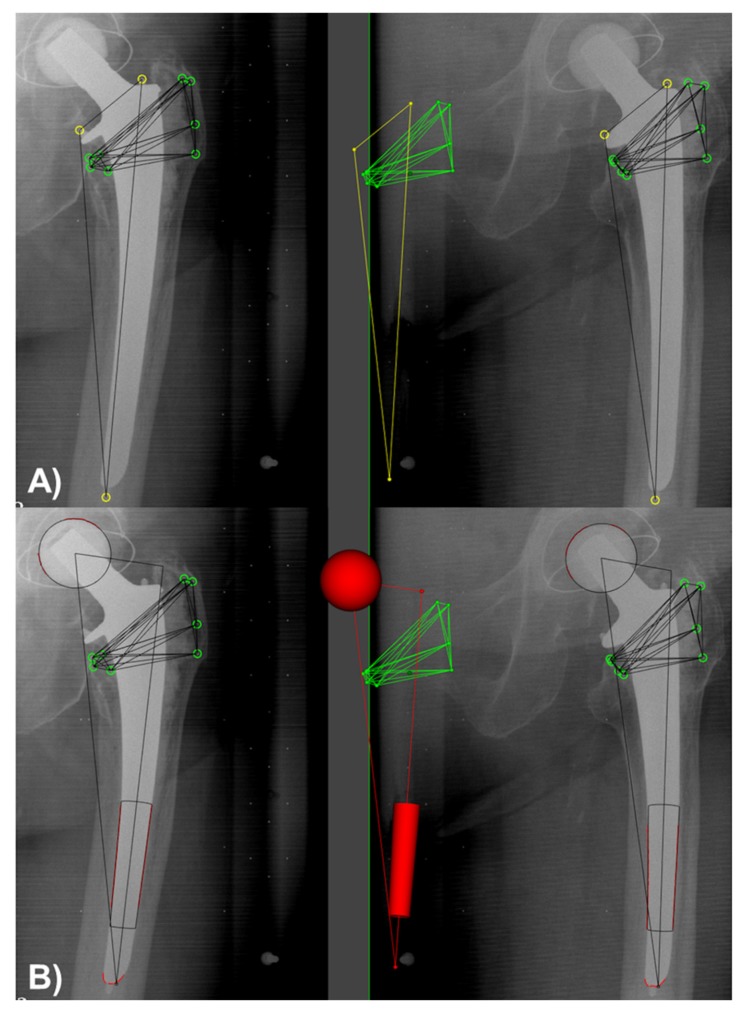
Same pair of Roentgen stereophotogrammetric analysis (RSA) radiographs analyzed by the both RSA approaches (**A**) marker-based and (**B**) model-based RSA (EGS). Bone markers are represent as green circles within the left and right X-ray of RSA image pair as well as green dots in their resulting 3D position (x, y, z coordinate) within the RSA analysis scene. Additional attached implant markers for marker-based RSA approach are illustrated as yellow circles and dots, respectively. The EGS model for model-based RSA approach is represented as red contours at the ball head, with the both X-rays and as red cylinder, sphere with the RSA analysis scene.

**Figure 2 materials-13-01543-f002:**
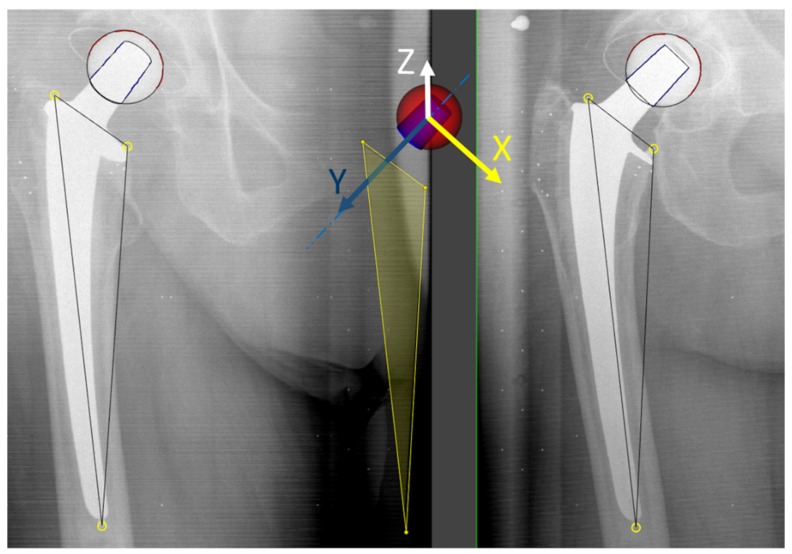
Coordinate system based on the long axis of the taper (by fitting a cone model to the contour of taper) and the three implant markers (yellow circles) to present head–taper migrations. The Y-axis was set on the long axis of the taper. The X-axis was orthogonal to Y-axis and parallel to a plane across three implant markers (yellow plane). The Z-axis of the new coordinate system was orthogonal to the X- and Y-axes.

**Figure 3 materials-13-01543-f003:**
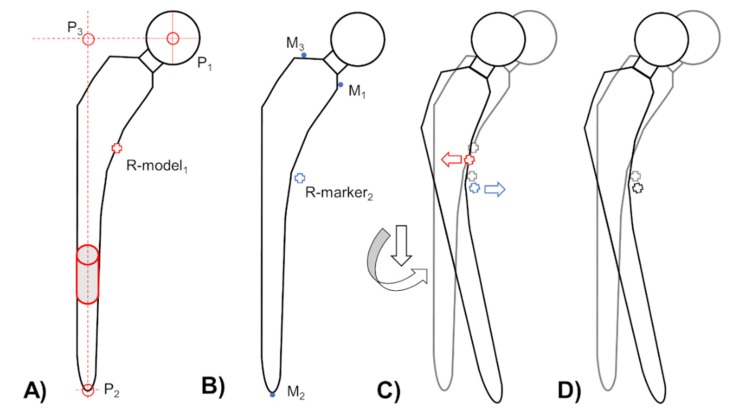
Resulting reference points for migration measurement of the same femoral stem design using (**A**) model-based RSA (red cross; R-model_1_) and (**B**) marker-based RSA (blue cross, R-marker_2_) approach. (**C**) If a stem subsidence with a combined rotation (black arrows) around from the anterior–posterior direction occurs, each reference point of the marker (blue cross and arrow) and model-based RSA (red cross and arrow) indicate a different directions of migration (relative to grey crosses). (**D**) To compare in vivo migration of the same RSA image pairs, a reference point correction is necessary. To enable this mentioned comparison of migration results the reference point of model-based RSA (EGS) was corrected to the reference point of marker-based RSA.

**Figure 4 materials-13-01543-f004:**
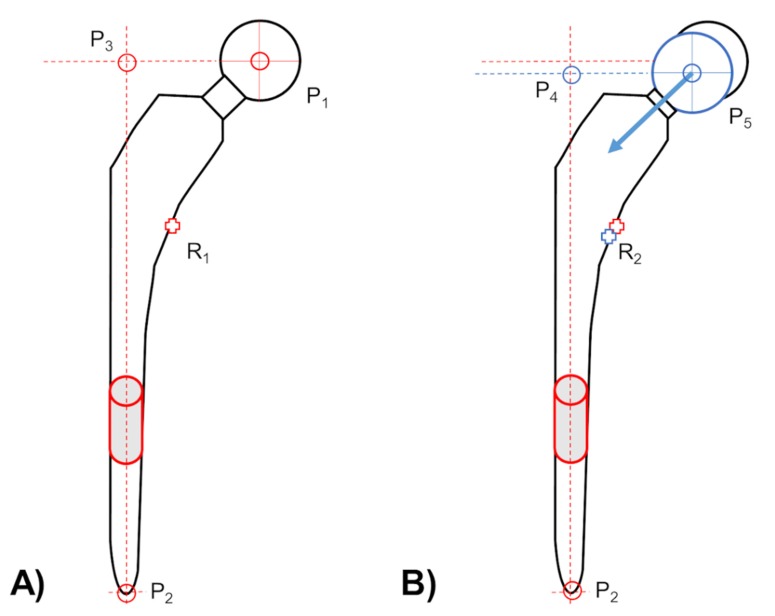
(**A**) An EGS model of a femoral stem component, consist out of three virtual markers (red circles). First marker indicates the center of the femoral ball head (P_1_). Second marker indicates the lowest point of the tip (P_2_). Last virtual marker (P_3_) is defined by the intersection of the horizontal extension line of point P_1_ and the perpendicular (vertical) line, resulting out of the point P_2_ and longitudinal axis of applied EGS cylinder model (gray cylinder with red contour). This determination of the femoral stem as rigid body with three virtual markers requires a stable head–taper connection. (**B**) Therefore, the three virtual markers are no longer a rigid body if the head center marker (P_1_ to P_5_) and its projection marker (P_3_ to P_4_) may migrate relative to the tip marker (P_2_) during the follow-up time period, and afterwards introduce a bias in the applied reference point (changing from R_1_ to R_2_) measurement with EGS stem model.

**Figure 5 materials-13-01543-f005:**
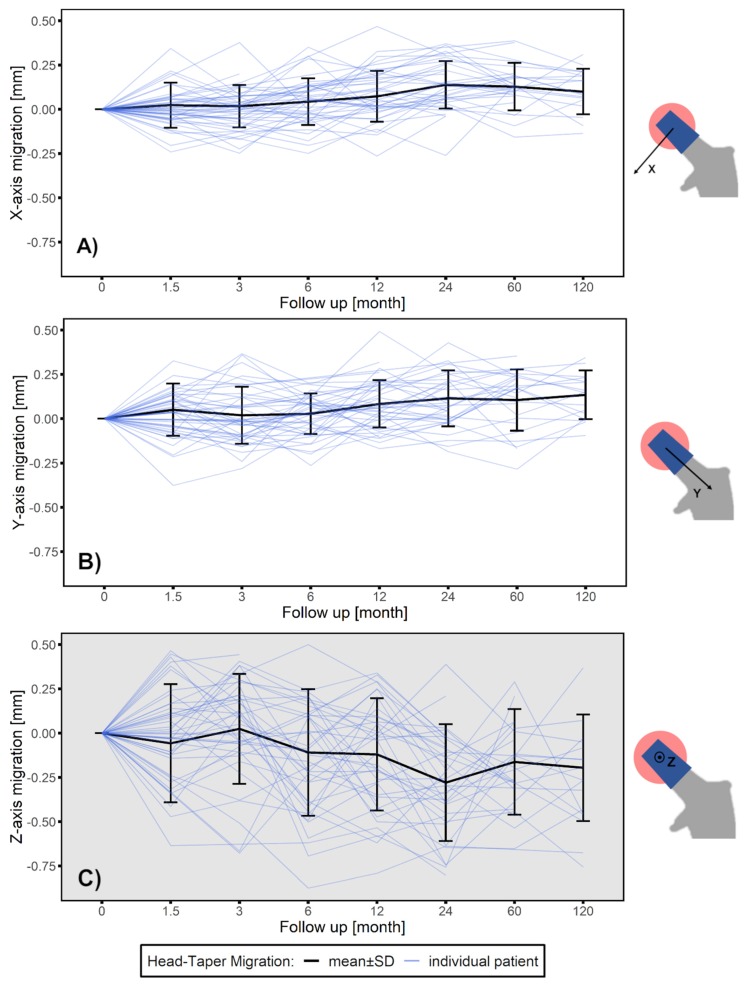
Head–taper migration depicted as mean ± SD (black line) for a ten-year follow-up time period along (**A**) medial-lateral direction perpendicular to the taper (x-axis translation), (**B**) cranial–caudal direction along the taper longitudinal axis (y-axis translation) and (**C**) anterior–posterior direction perpendicular to the taper (z-axis translation). Blue lines indicate each individual data set. Gray background indicate out-of-plane measurement.

**Figure 6 materials-13-01543-f006:**
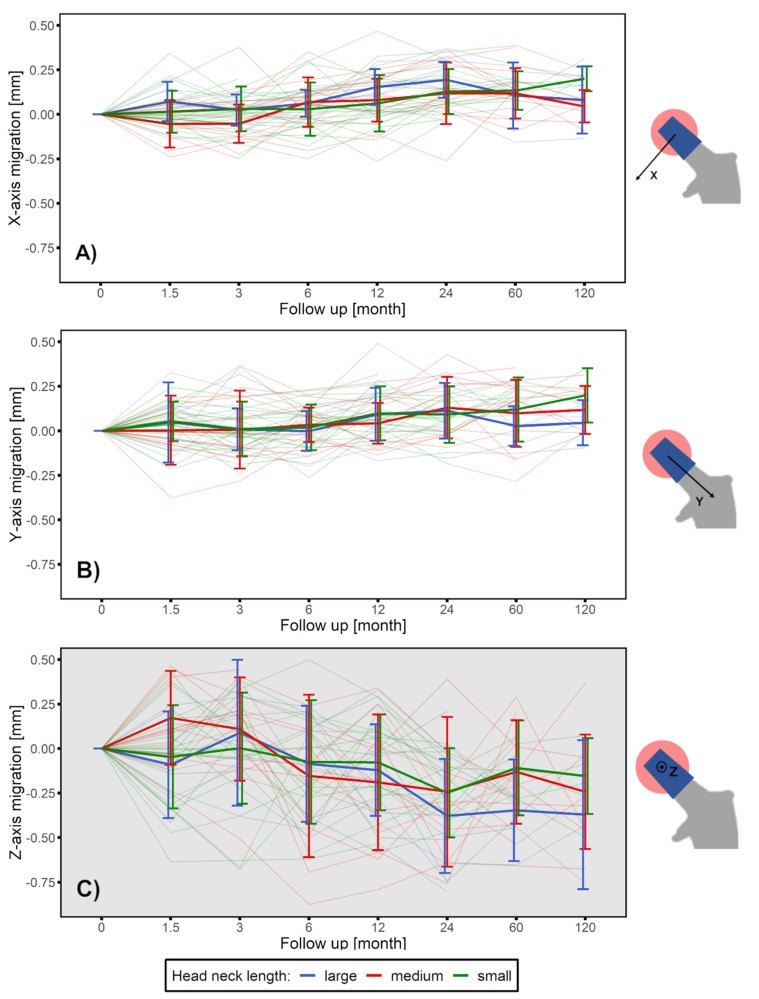
Head–taper migration depicted as mean ± SD for a ten-year follow-up time period along (**A**) medio-lateral direction perpendicular to the taper (x-axis translation), (**B**) cranio–caudal direction along the taper longitudinal axis (y-axis translation) and (**C**) anterior–posterior direction perpendicular to the taper (z-axis translation), grouped by head–neck length: small (green line), medium (red line) and large (blue line). Gray background indicate out-of-plane measurement.

**Figure 7 materials-13-01543-f007:**
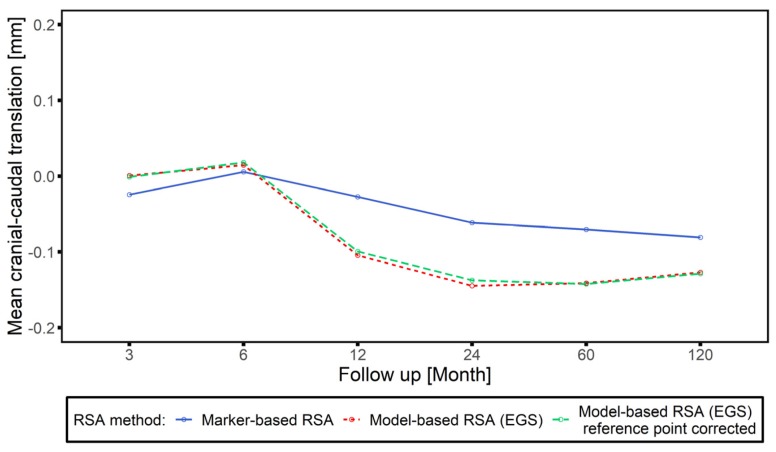
Cranial–caudal translation along follow-up time period analyzed by marker-based RSA (blue solid line), model-based RSA (EGS) (red dotted line) and model-based RSA (EGS) after reference point correction (green dashed line).

**Figure 8 materials-13-01543-f008:**
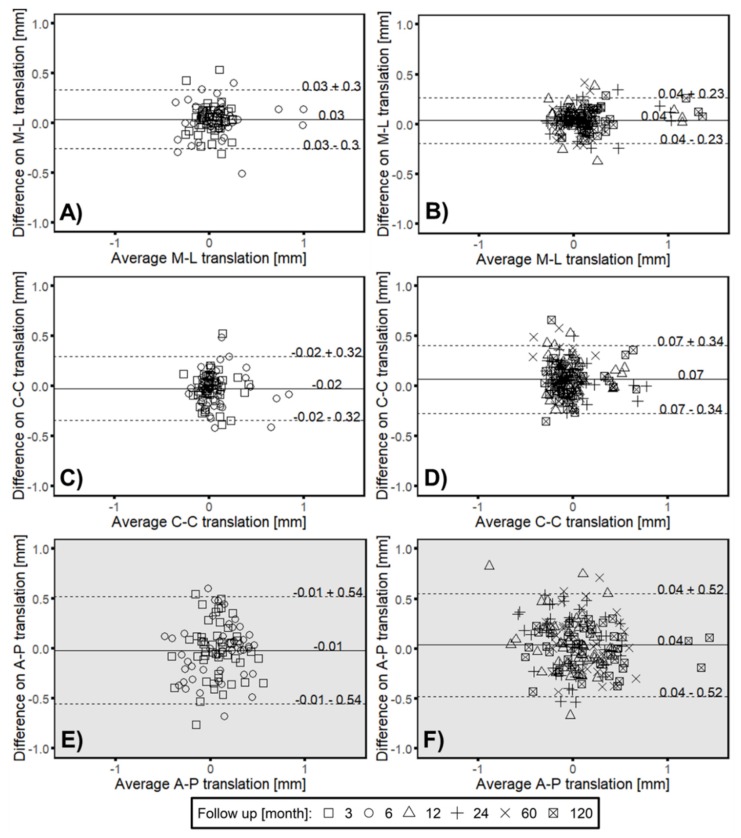
Bland–Altman plots presenting of the calculated average and difference between marker-based RSA and model-based RSA (EGS) for translational migration measurements along medial-lateral (ML), cranial–caudal (C-C) and anterior–posterior direction (AP). Solid line indicates the mean difference with LoA (dashed line) for (**A**,**C**,**E**) 3 (square) and 6 month follow up (circle) as well as for (**B**,**D**,**F**) 12 (triangle), 24 (plus), 60 (cross) and 120 (square with cross) month follow up respectively. Gray background indicate out-of-plane measurement. Mean difference equal to 0 mm: (**A**): *p* < 0.05; (**B**): *p* < 0.05; (**C**): *p* = 0.20; (**D**): *p* < 0.05; (**E**): *p* = 0.60; (**F**): *p* < 0.05.

**Figure 9 materials-13-01543-f009:**
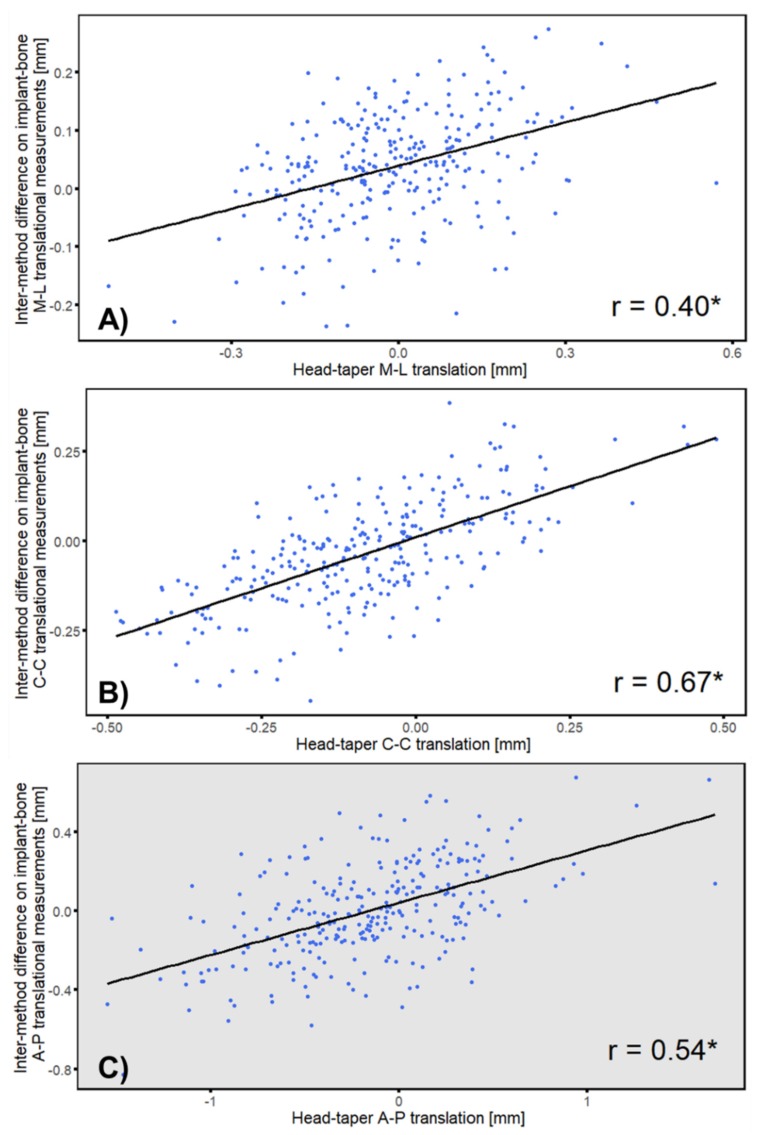
Correlation of intermethod difference and head–taper translation along (**A**) medial-lateral (**B**) cranial–caudal and (**C**): anterior–posterior direction. * *p* ≤ 0.05. Gray background indicate out-of-plane measurement.

**Table 1 materials-13-01543-t001:** Stem-taper geometry and topography.

Parameter	Value
Taper angle	5.611 ± 0.004 deg
Ra	3.58 ± 0.06 µm
Rz	14.26 ± 0.29 µm
RSm	122.93 ± 0.20 µm
Rp	7.50 ± 0.19 µm
Rk	13.90 ± 0.34 µm

**Table 2 materials-13-01543-t002:** Migration along the taper axis.

Head–Taper Junction	0 > 2 kN	2 > 4 kN	Total
#1	160 µm	105 µm	265 µm
#2	155 µm	95 µm	250 µm
#3	160 µm	125 µm	285 µm
#4	173 µm	100 µm	273 µm
#5	160 µm	105 µm	265 µm

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
