# Peer review of "Model-Based Roentgen Stereophotogrammetric Analysis to Monitor the Head–Taper Junction in Total Hip Arthroplasty in Vivo—And They Do Move"

_materials, 2020, doi:10.3390/ma13071543_

Round 1

Reviewer 1 Report

The authors developed a model-based RSA method for the study of the THA head-taper movement in vivo, and they observed substantial subsidence. The topic is important and the study has been carried out carefully. The present reviewer recommends publication provided that the following comments are responded to.

Discussion. The authors should add more discussion about the possible clinical significance of the subsidence that they observed. For instance, could it increase the fracture risk of the ceramic head? Wear of the male taper?

Line 345. “mechanical saturation” is a vague expression and should be rewritten. Are hoop stresses, accommodation of surfaces, slow crack growth, etc. involved for instance? Please clarify.

There are numerous grammatical errors. Therefore the check of English must be done.

Reviewer 2 Report

The study investigates movement at the head-taper junction in THR, an issue that so far has been widely neglected . The coupling is limited to a metal- ceramic coupling. The methods chosen appear sound and are well described. Results are interesting but appear not impressive in the present form. I believe the manuscript is too long for the presented results. The measured movement per se would not be clinically relevant, however, it may be concluded, that the change of the distances inevitably must be related with some abrasion of the material, presumably the metal part. The resulting debris may cause several undesired effects, like local inflammation, resorption of bone or even elevated serum levels. Trunnionosis is a frequently discussed issue in THR; the presented study might serve a valuable tool in investigating cases with suspicion of  one of the conditions. I believe this aspect is not sufficiently covered.

Reviewer 3 Report

Line 81: change "proof" to "prove"

Line 100: Change "Patients were stored" to "patients were positioned"

Please check tense throughout manuscript, make sure past tense is used consistently when describing Methods and Results.

Line 137: replace "had been realized with" with "matched"

Line 149: change "proof" to "prove"

Line 210: change "RGS" to "EGS"

Figure 6: put colour key as a legend on the plots

Line 319: Change "proofing" to "proving"

Line 372: change "proof" to "prove"

Round 2

Reviewer 2 Report

The revised manuscript now also refers to possible clinical relevance and such is much more interesting to the readers. Still I believe some more shortening may be possible in my opinion.

… model-based RSA method was NOT introduced at the beginning of 20th century but the 21st century.
